# The Development of a Skin Image Analysis Tool by Using Machine Learning Algorithms

**Perry Xiao** [1,*], **Xu Zhang** [2], **Wei Pan** [3], **Xiang Ou** [4], **Christos Bontozoglou** [1], **Elena Chirikhina** [1] **and Daqing Chen** [1]

[1] School of Engineering, London South Bank University, 103 Borough Road, London SE1 0AA, UK; bontozoc@lsbu.ac.uk (C.B.); chirikhe@lsbu.ac.uk (E.C.); chend@lsbu.ac.uk (D.C.)

[2] Auckland Tongji Medical & Rehabilitation Equipment Research Centre, Tongjing Zhejiang College, JiaXing 314000, China; zhangx12@tjzj.edu.cn

[3] School of Computer Science and Engineering, Changshu Institute of Technology, Changshu 215500, China; pvv1224@163.com

[4] College of Communication Engineering, Chongqing University, Chongqing 400044, China; ouxiang@cqu.edu.cn

\* Correspondence: xiaop@lsbu.ac.uk; Tel.: +44-20-7815-7569; Fax: +44-20-7815-7561

**Abstract:** We present our latest research work on the development of a skin image analysis tool by using machine-learning algorithms. Skin imaging is very import in skin research. Over the years, we have used and developed different types of skin imaging techniques. As the number of skin images and the type of skin images increase, there is a need of a dedicated skin image analysis tool. In this paper, we report the development of such software tool by using the latest MATLAB App Designer. It is simple, user friendly and yet powerful. We intend to make it available on GitHub, so that others can benefit from the software. This is an ongoing project; we are reporting here what we have achieved so far, and more functions will be added to the software in the future.

**Keywords:** graphic user interface; deep learning; machine learning; capacitive imaging; skin image analysis

## 1. Introduction

Skin imaging is very import in skin research. To date, there are many popular skin imaging technologies. Dermatoscopy is probably the most widely used skin imaging technology and is commonly used to examine skin lesions with a dermatoscope [1,2]. A dermatoscope typically consists of a magnifier, a light source (polarized or non-polarized) and a transparent plate. Skin photos and videos can be recorded by using a digital camera. DermLite is a small dermatoscope that can be attached to a smart phone. It claims to be the world's best-selling dermatoscope [3]. DermLite II MultiSpectral is another pocket size dermatoscope, it is button activated, with cross-polarized 4-color imaging [4]. Nerasolutions provides several imaging devices, not only for skin imaging, but also for hair imaging [5]. Visia Skin Analysis System takes multiple photos of face and analyzes the melanin distributions, the texture, the wrinkle, the pores, etc. [6]. EPISCAN I-200 is a high resolution ultrasound imaging system that can take skin images using ultrasound up to 50 MHz [7]. Last not the least, Epsilon permittivity imaging system is a capacitive image technology developed in our research group [8–13]. It is based on dielectric constant measurement principle, and has been used for skin hydration imaging, skin solvent penetration imaging, as well as hair and nail imaging. With all these skin imaging technologies, there is a genuine need for a general purpose, user friendly and yet powerful skin software analysis tool. In this paper, we present our latest work on the development of a

skin image analysis tool based on machine-learning algorithms. We will first present the design and the structure of the skin software analysis tool, then present the results and discussions and finally, the future works.

## 2. Materials and Methods

### 2.1. Front-End GUI

The front-end GUI (graphical user interface) is designed and developed by using MATLAB App Designer, which is relatively new and provides much better user interface components and a better integrated development environment than the previous MATLAB GUIDE. With MATLAB App Designer, you can easily design and develop a user friendly software interface. It offers a grid layout manager to organize the user interface and automatic reflow options to make the app detect and respond to changes in screen size. The coding is also a lot simpler, with machine generated code automatically grayed out. Figure 1 shows the GUI of the skin image analysis software tool. The left panel shows the original image and analyzed results. The right panel contains several tabs with each tab is dedicated to different functions. Apart from reading from image files, it can also get image frames from webcams and other USB based video devices. This is done by using "MATLAB Support Package for USB Webcams" Add-On module which has shown significant performance improvement than the existing video input function from the image acquisition toolbox.

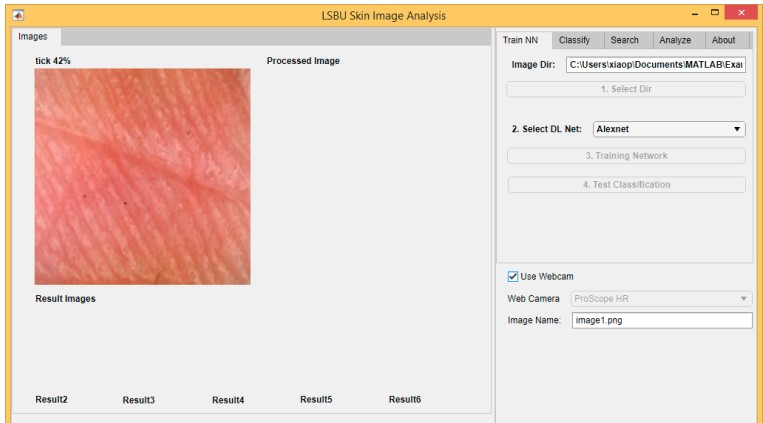

**Figure 1.** The GUI (graphical user interface) of the skin image analysis tool.

### 2.2. Back-End MATLAB Files

The back-end of the program is implemented in a number of MATLAB functions saved as standard M-files. MATLAB functions, also called modules or subroutines in other programming languages, accept input arguments and return output arguments. They operate on variables within their own workspace, different from the main program. Table 1 shows the structure of the skin analysis software tool.

**Table 1.** Structure of the skin analysis tool.

| Files | Comments |
| --- | --- |
| PX_DL_GUI.mlapp | Main GUI file |
| AlexnetTraining.m | AlexNet training function |
| AlexnetClassify.m | AlexNet classification function |
| GoogLeNetTraining.m | GoogLeNet training function |
| GoogLeNetClassify.m | GoogLeNet classification function |
| Gabor_Calculate.m | Gabor wavelet transform calculation function |
| Gabor_Search.m | Gabor wavelet transform search function |
| … … | … … |

*2.3. Functionalities*

The main purpose of this work is to develop a general purpose skin image analysis software with many useful skin image analysis functions. It is also designed to be flexible, so more new functions can be added in the future.

The current implemented functions are image classifications [8,9,13], skin texture analysis and skin image retrieval by using Gabor wavelet transform, PCA (principle components analysis) and GLCM (gray-level co-occurrence matrix) [10,11], as well as skin live image analysis.

## 3. Results and Discussions

*3.1. Skin Image Classification*

Skin image classification is the most important function of the skin analysis tool. With skin image classification, we would be able to differentiate normal skin from damaged skin, healthy skin from diseased skin. It can also potentially be used for identifying different types of skin diseases and skin cancers. Skin image classification is achieved through transfer learning by using pre-trained deep-learning neural networks such as AlexNet [14], GoogLeNet [15], VGG19 [16] and ResNet101 [17]. The performance of image classification is tested on four different types of images, as shown in Figures 2–5. All the images are randomly divided into training set (70%) and test set (30%) and automatically resized to $224 \times 224 \times 3$ (except for AlexNet, $227 \times 227 \times 3$) before training.

The grayscale images also automatically converted to color images prior to training. During the training, it uses stochastic gradient descent with momentum, the initial learning rate is 10ˆ-4, mini batch size is 5, max epochs is 6, and it also shuffles training between every epoch. The trainings are done on a standard desktop computer with Intel® Core™ i7-3770 CPU @3.4 GHz, 8 cores, 16 GB RAM and Windows 8.1 operating system.

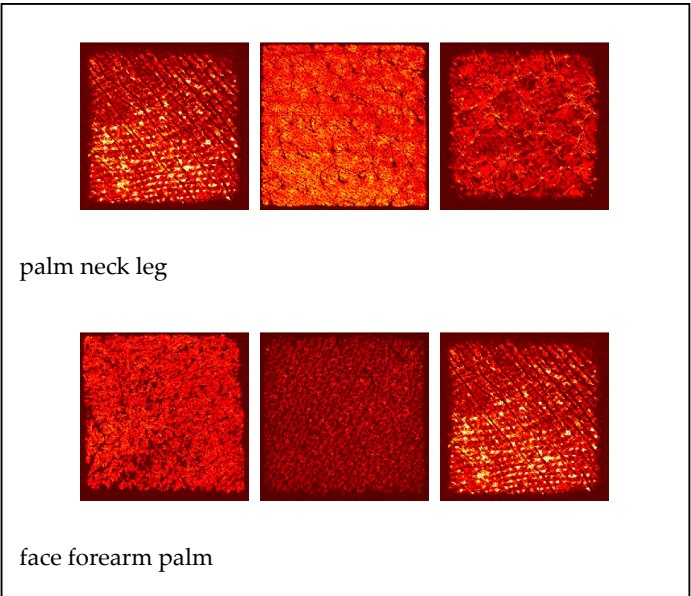

palm neck leg

face forearm palm

**Figure 2.** Sample skin capacitive images from different skin sites of a healthy volunteer. There are total 90 images divided into six categories, palm, neck, leg, face, forearm and palm.

Figure 2 shows the skin capacitive images acquired by using our Epsilon permittivity imaging system [8,9]. There are total 90 images divided into six categories, palm, neck, leg, face, forearm and palm. The image classification results are shown in the first row of Table 2. AlexNet has the shortest training time and VGG19 has the longest training time. The classification overall accuracy is good, except VGG19 and GoogLeNet gives an impressive 100% classification accuracy.

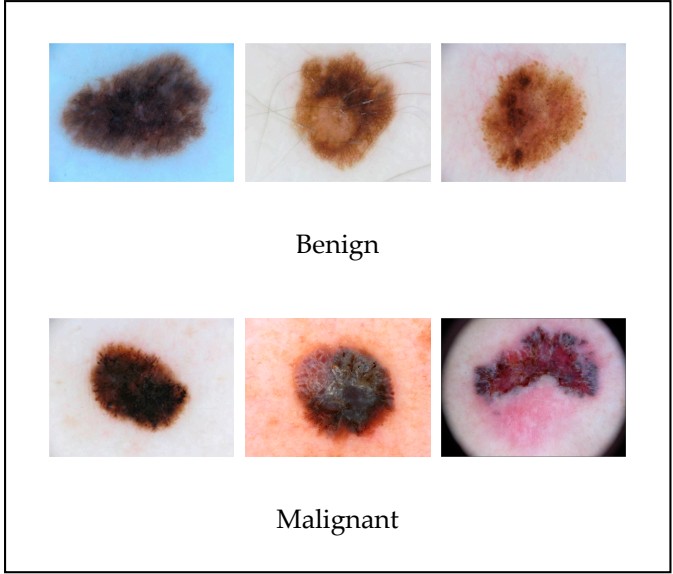

**Figure 3.** Sample skin cancer images from ISIC database [18]. There are total 600 images divided into two categories, Benign and Malignant.

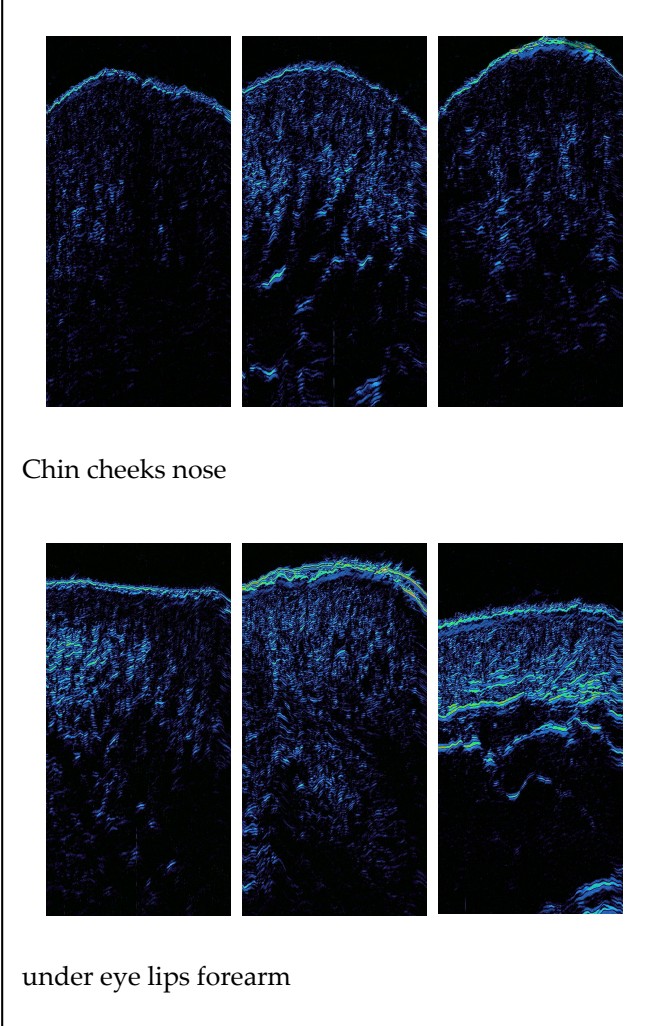

**Figure 4.** Sample skin ultrasound images from different skin sites of a healthy volunteer. There are total 610 images divided into six categories, chin, cheeks, nose, under eye, lips and forearm.

Figure 3 shows the skin cancer images from ISIC database [18]. These are standard digital photo images. There are totally 600 images divided into two categories: Benign and Malignant. The image classification results are shown in the second row of Table 2. AlexNet again has the shortest training time, and VGG19 has the longest. GoogLeNet and ResNet101 have the best classification accuracy.

Figure 4 shows the skin ultrasound images acquired by using EPISCAN I-200 ultrasound instrument. There are totally 610 images measured from six different skin sites, chin, cheeks, nose, under eye, lips and forearm. The image classification results are shown in the third row of Table 2. AlexNet again has the shortest training time and VGG19 the longest. ResNet101 has the best classification accuracy. The accuracy is slightly lower than lower than previous two types of skin images, this is understandable, as skin ultrasound images have higher similarities.

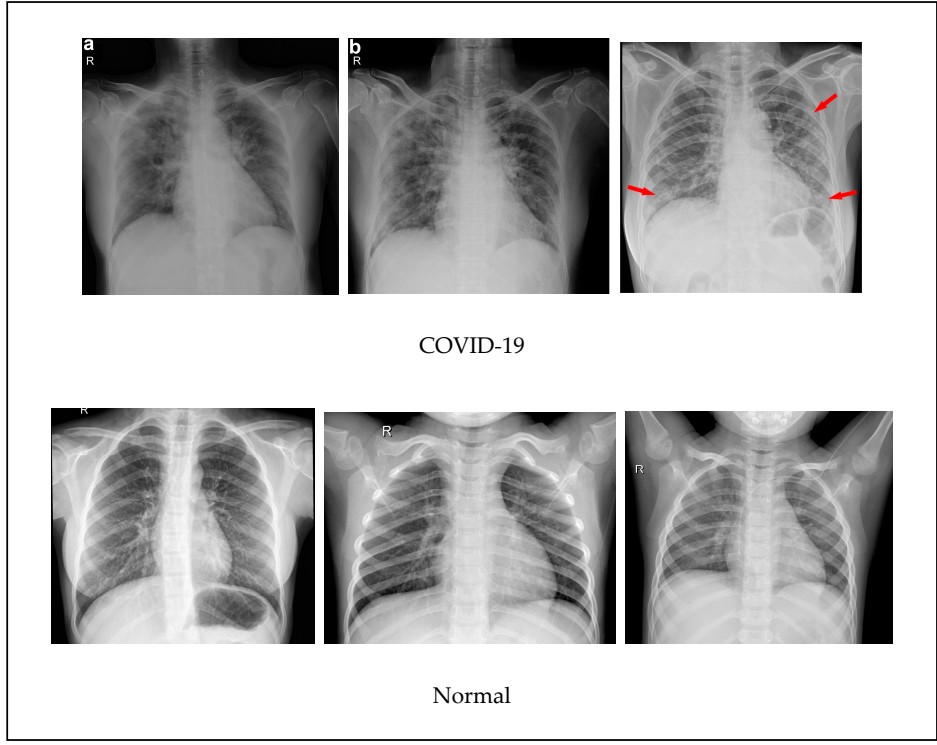

COVID-19

Normal

**Figure 5.** Chest X-ray images from University of Montreal [19] and Kaggle [20]. There are total 50 images divided into two categories, COVID-19 and normal.

Currently, the coronavirus COVID-19 has become a global pandemic and is affecting more than 21 million people in more than 200 countries and territories around the world. We wanted to see if our software tool can effectively work on COVID-19 chest X-ray images, as shown in Figure 5. The COVID-19 chest X-ray images are from a research group in University of Montreal [19] and the normal chest X-ray images are from Kaggle [20]. There are totally 50 images divided into two categories: COVID-19 and normal. The image classification results are shown in the third row of Table 2. AlexNet, not surprisingly, has the shortest training time and VGG19 the longest. The overall classification accuracy is high (>85%) and ResNet101 has an impressive 100% classification accuracy.

In summary, AlexNet is consistently the fastest in image classification training, while VGG19 is the slowest. The ResNet101 has the highest overall classification accuracy and VGG19 has the lowest overall classification accuracy. For skin capacitive image classification, GoogLeNet is the best. For other image classification, e.g., skin cancer images, skin ultrasound images and Chest X-ray images, ResNet101 is the best.

**Table 2.** Image classification results.

| Image Types | Table Column Head | | |
| --- | --- | --- | --- |
| | **Models** | **Training Time** | **Accuracy** |
| Capacitive skin images | AlexNet | 1 min 6 s | 83.33% |
| | GoogLeNet | 1 min 39 s | 100.00% |
| | VGG19 | 12 min 5 s | 58.33% |
| | ResNet101 | 9 min 12 s | 87.50% |
| Skin cancer images | AlexNet | 8 min 37 s | 73.89% |
| | GoogLeNet | 13 min 23 s | 77.78% |
| | VGG19 | 91 min 15 s | 75.00% |
| | ResNet101 | 66 min 40 s | 77.78% |
| Skin ultrasound images | AlexNet | 4 min 56 s | 69.44% |
| | GoogLeNet | 10 min 51 s | 56.67% |
| | VGG19 | 71 min 59 s | 60.56% |
| | ResNet101 | 63 min 3 s | 70.56% |
| Chest X-ray images | AlexNet | 43 s | 92.86% |
| | GoogLeNet | 1 min 30 s | 92.85% |
| | VGG19 | 6 min 44 s | 85.71% |
| | ResNet101 | 4 min 42 s | 100.00% |

The most significant benefit of skin image classification is that users would be able to use the skin image analysis tool with a webcam camera to take a picture of a skin lesion, the software will be able to tell whether it is benign or malignant. Users can also train the neural networks on their own skin image data, to recognize different types of skin diseases/cancers. This could be a potential low cost skin diseases/cancer diagnosis device.

The classification accuracy depends on the number and the quality of skin images, as well as the deep-learning neural networks. To further improve the accuracy, we will need train the neural networks on larger number of skin images and better quality of skin images. Currently we only have tested four different deep-learning neural networks, in the future, we will implement more neural networks. In addition, all the neural networks are trained on a set of default optimized values, next step is to allow users to specify their own training parameter values.

*3.2. Skin Texture Analysis and Image Retrieval*

Skin surface is not even, it is rich with patterns and lines. Our previous studies show that we can analyze the skin texture by using Gabor wavelet transform, PCA (principle components analysis) and GLCM (gray-level co-occurrence matrix). With Gabor wavelet transform, we can get skin features vectors on color, shape and texture [10]. With GLCM (gray-level co-occurrence matrix), we can get four GLCM feature vectors, e.g., angular second moment (ASM), entropy (ENT), contrast (CON) and correlation (COR). The results showed that the angular second moment increases as age increases and entropy decreases as age increases [11]. We can also analyze the skin micro-relief, the results showed that the change in the intensity of primary and secondary lines during arm extension is increasing against chronological age, while the average number of closed polygons per square mm is decreasing against chronological age [12].

Another useful usage of skin texture analysis is image retrieval or image searching. This is particularly suitable for skin images, where similarities are much higher than normal images. Figure 6a shows the skin cancer image searching results using Gabor wavelet transform. As we use the same image database for training and searching, the best matching result is the query image itself. The second to sixth best matching results are also from the same category of the query image, i.e., benign images. Figure 6b–d show the image searching results using Gabor wavelet transform on skin capacitive images, skin ultrasound images and chest X-ray images. Similar to Figure 6a, all the best matching results are from the same category of the query image.

Table 3 shows the image searching results of different algorithms on different types of images. Calculation time is how long each algorithm takes to calculate the feature vectors of all the images and built a database of the results. Searching time is how long each algorithm takes to search the database and find the six best matching results. The error rate is calculated as how many images in the best six matching results are in wrong category. Figure 6 shows the best scenario of Gabor wavelet transform searching results, the error rates are 0 out of 6, while Table 3 shows the average error rates. The overall results show that Gabor wavelet transform general takes longer time to calculate and to search, while PCA is the shortest for calculation and GLCM is the shortest for searching. Although Gabor wavelet transform is slower compared with other image retrieval techniques, such as PCA and GLCM, Gabor wavelet transform is the best for retrieving images. This agrees well our previous studies on skin capacitive contact images and digital facial images [10]. All the algorithms are running on a set of optimized default values, in the future, we will allow users to specify their own parameter values.

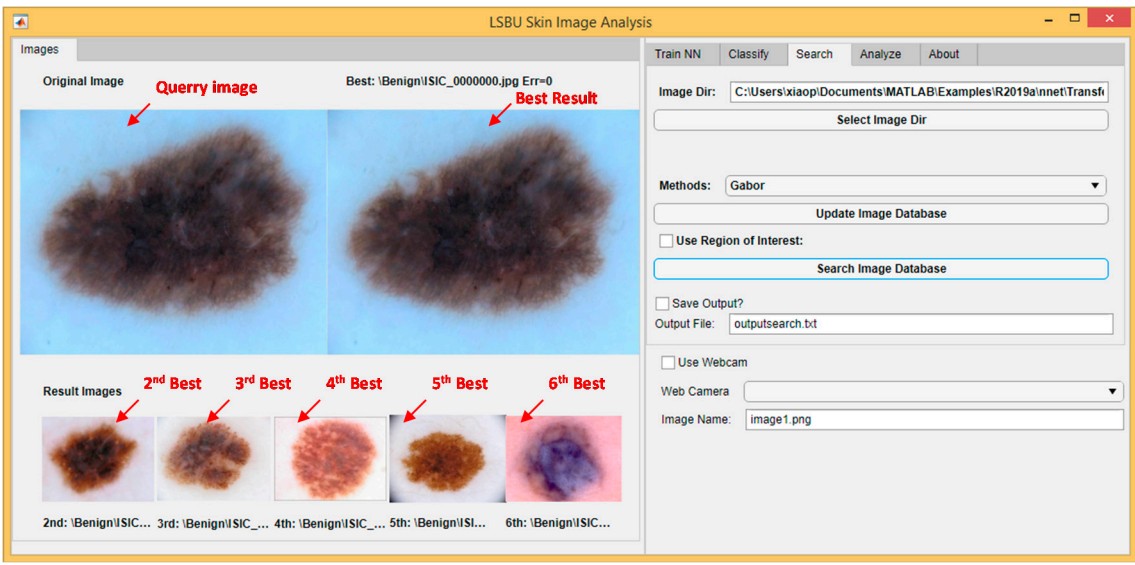

(**a**)

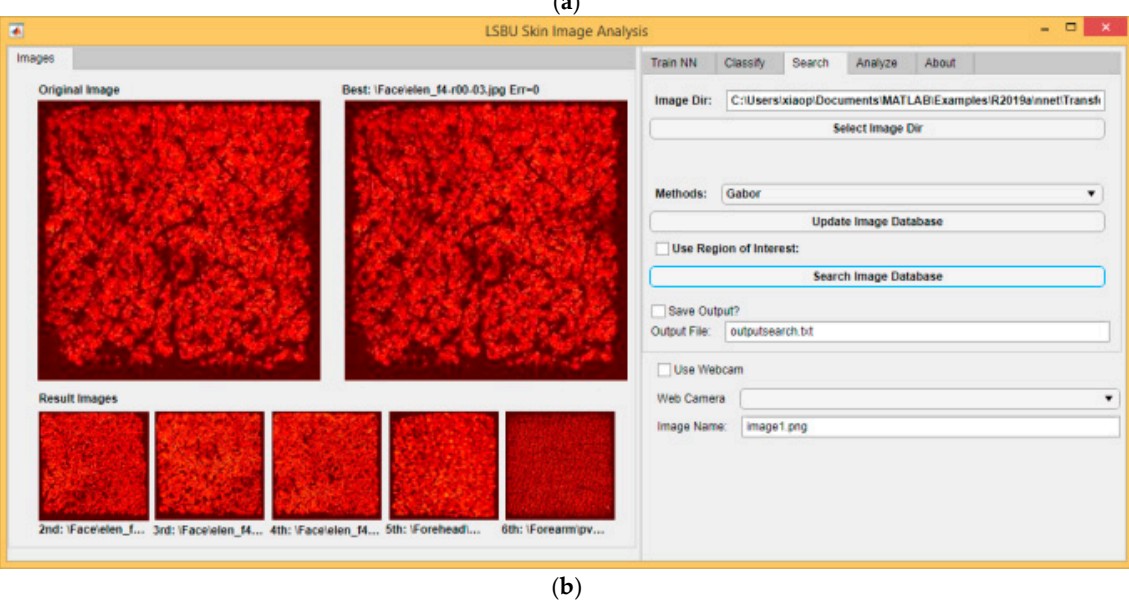

(**b**)

**Figure 6.** *Cont.*

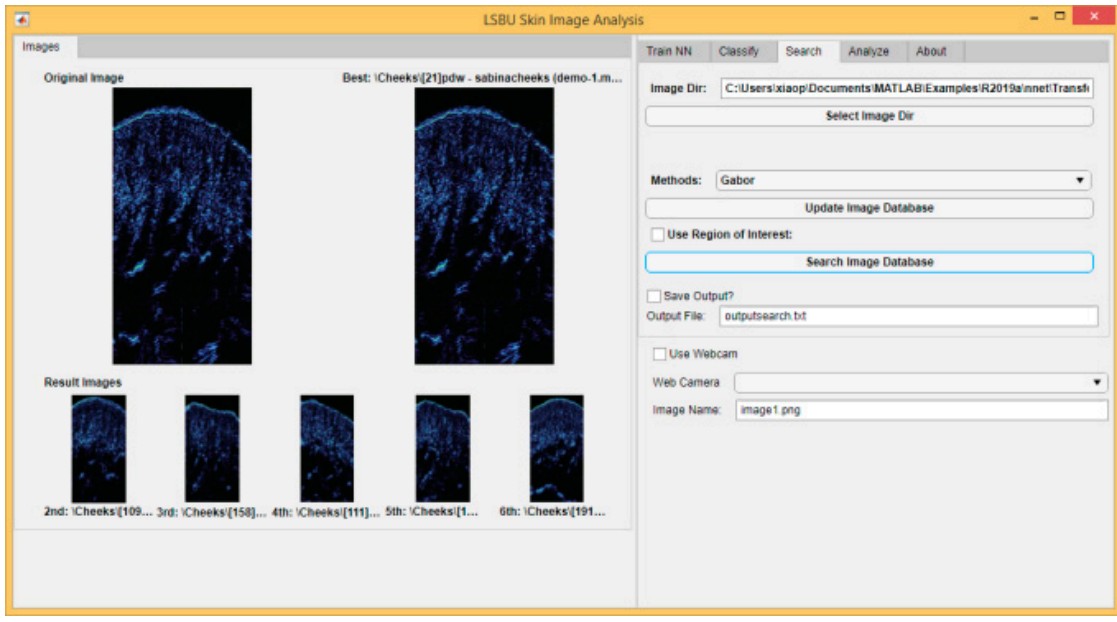

(**c**)

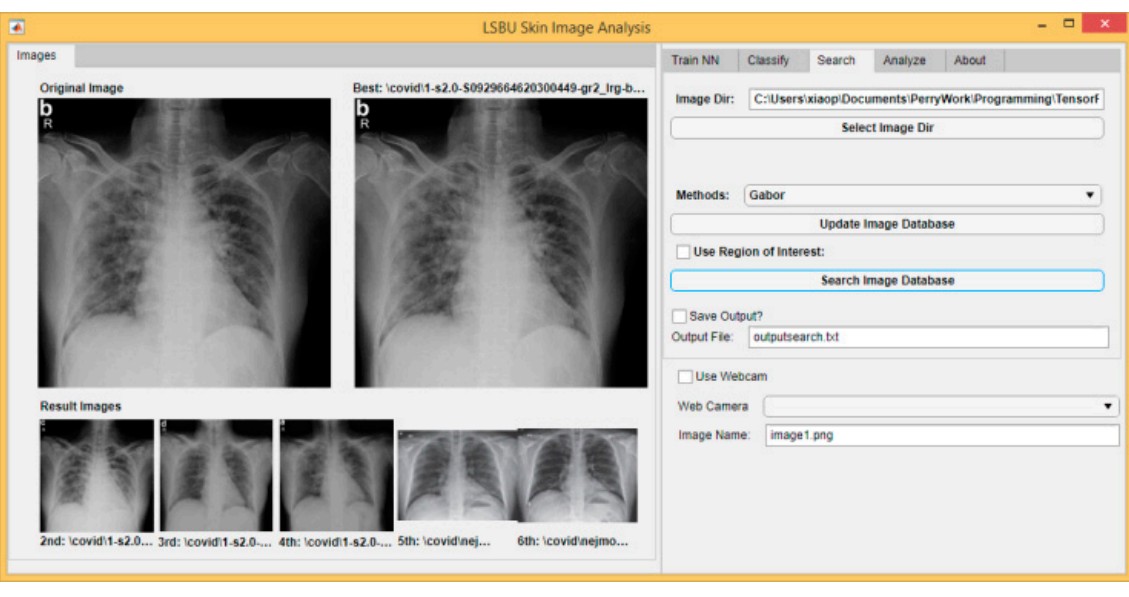

(**d**)

**Figure 6.** Image searching results using Gabor wavelet transform on different type of images. (**a**) skin capacitive images; (**b**) skin cancer images; (**c**) skin ultrasound images; (**d**) chest X-ray images.

**Table 3.** Image searching results.

| Image Types | Table Column Head | | | |
|---|---|---|---|---|
| | Algorithms | Calculation Time | Searching Time | Error Rate |
| Capacitive skin images | Gabor | 153 s | 1059 ms | 1/6 |
| | PCA | 1.6 s | 221 ms | 3/6 |
| | GLCM | 1.1 s | 25 ms | 2/6 |
| Skin cancer images | Gabor | 370 s | 2834 ms | 0/6 |
| | PCA | 54 s | 1560 ms | 2/6 |
| | GLCM | 482 s | 698 ms | 2/6 |

**Table 3.** *Cont.*

| Image Types | Table Column Head | | | |
| --- | --- | --- | --- | --- |
| | **Algorithms** | **Calculation Time** | **Searching Time** | **Error Rate** |
| Skin ultrasound images | Gabor | 324 s | 2743 ms | 2/6 |
| | PCA | 29 s | 2409 ms | 3/6 |
| | GLCM | 118 s | 220 ms | 1/6 |
| Chest X-ray images | Gabor | 35 s | 813 ms | 0/6 |
| | PCA | 2.6 s | 162 ms | 2/6 |
| | GLCM | 30 s | 563 ms | 0/6 |

*3.3. Skin Live Image Analysis*

Apart from analyze image files, it can also analyze skin live images captured from webcams and other USB based video devices. The live image capture is achieved by using the latest MATLAB Add-On module called "MATLAB Support Package for USB Webcams". Figure 7 shows the live image analysis of facial skin using the ProScope HR with 50 times magnification [21]. The facial features such as skin texture, speckles and facial hair are clear to see. The software can divide the live image into three color channels, red, green and blue (RGB) and therefore perform the RGB calculation in real time. Figure 7a shows the original image and processed image (green channel only), Figure 7b shows the original image and processed image (red channel only). As we can see, green channel has more surface information, while red channel has more information on melanin (facial hair) and skin color.

Figure 7c shows the original image and the processed image, calculated by adding green channel × 0.7 with blue channel × 0.7, without red channel. As we can see, this enhances the surface texture information of the skin. Users can specify they own RGB channel calculations. Facial skin has enormous cosmetic values, this functionality could be very useful for analyzing skin facial features, such as facial hairs, smoothness, speckles, scars, wrinkles, all in real time! More functions will be added. The current version of the program is available as a MATLAB App Installer format on GitHub [22].

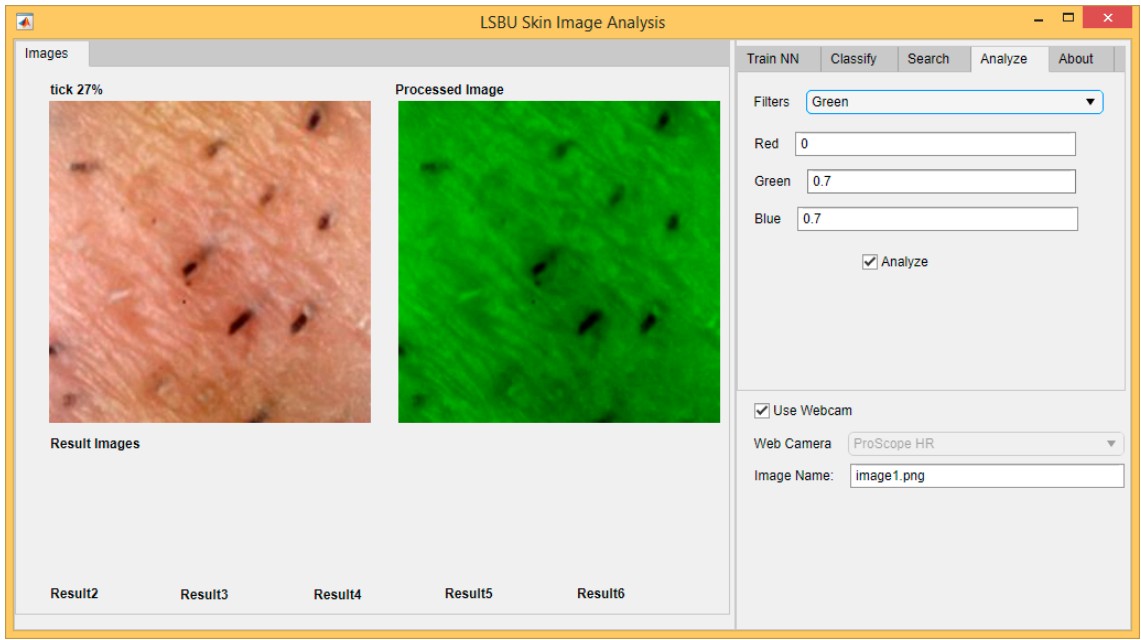

(**a**)

**Figure 7.** *Cont.*

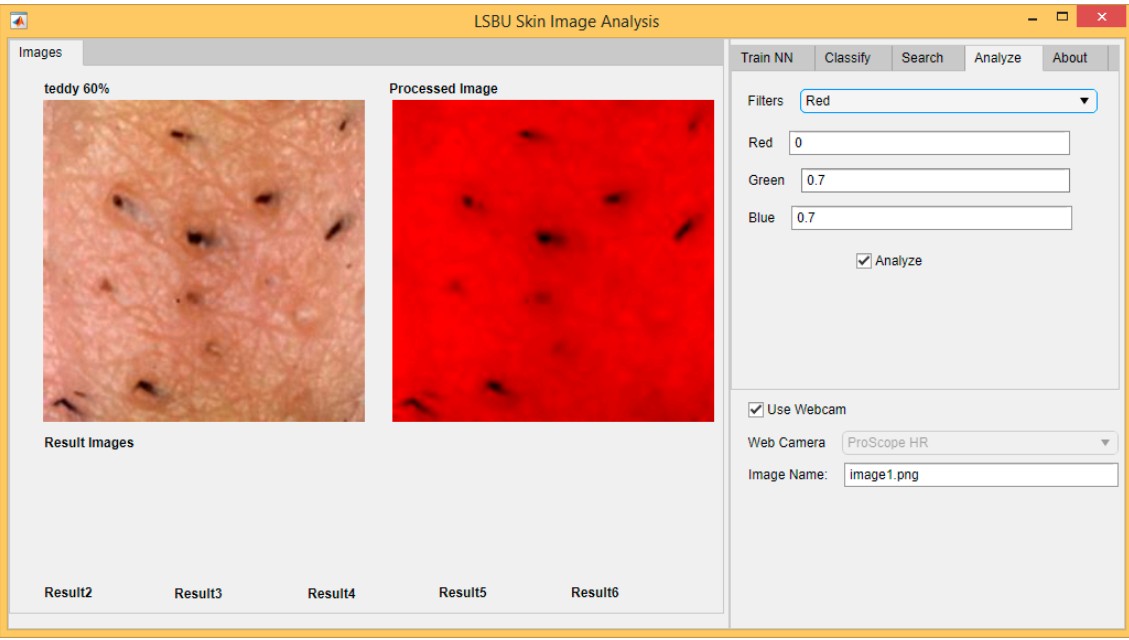

(**b**)

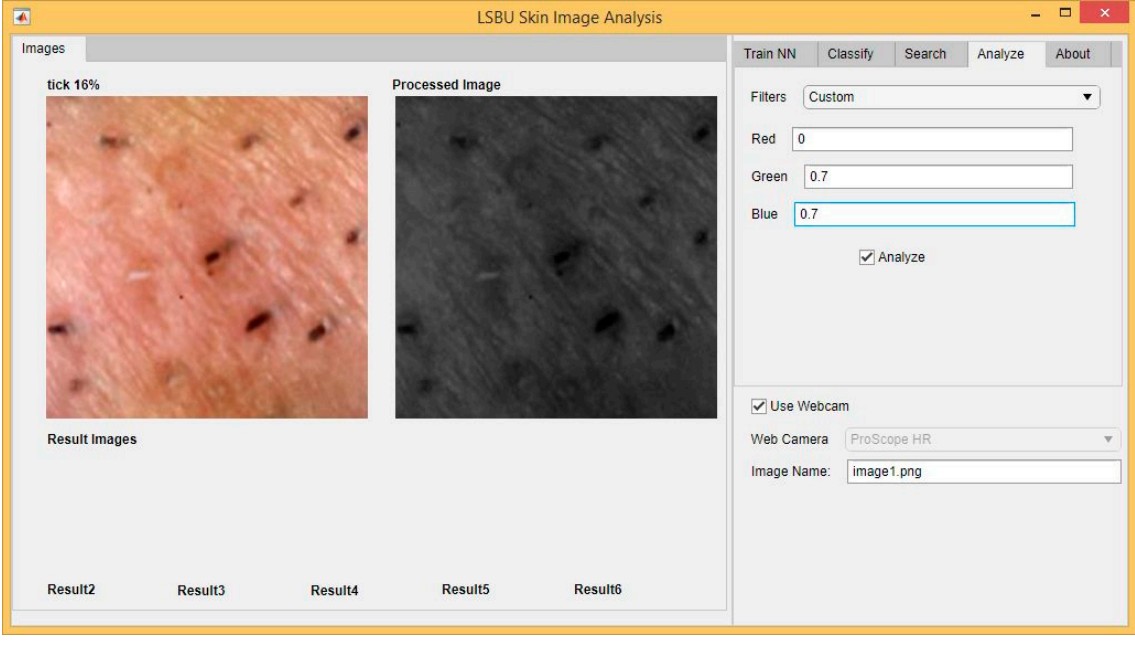

(**c**)

**Figure 7.** Live image analysis using the ProScope HR with 50 times magnification. (**a**) original image and processed image (green channel only); (**b**) original image and processed image (red channel only); (**c**) original image and the processed image (no red channel, green × 0.7 + blue × 0.7).

## 4. Conclusions

We have developed a skin image analysis tool based on machine-learning algorithms. It uses the latest MATLAB App Designer to create the graphic user interface on the front end and use MATLAB M-files as the backend. The current version provides functions such as skin image classifications, skin image retrieval and skin live image analysis. The skin image classification is done by using deep-learning neural networks such as AlexNet, GoogLeNet, VGG19 and ResNet101. The classification

results show that GoogLeNet is the best for skin capacitive image classifications, while ResNet101 is the best for skin cancer image, skin ultrasound image and chest X-ray image classifications. The biggest advantage of this skin image analysis tool is that users would be able to retrain the deep-learning neural networks on their own skin image data and therefore perform their own skin image classifications, such as recognizing different types of skin diseases. This could lead to a potential low cost skin disease/cancer diagnose device. The skin image retrieval is done by using Gabor wavelet transform, PCA and GLCM. The results show that Gabor wavelet transform has the overall best accuracy despite of being the slowest. In skin live image analysis, by manipulating the RGB values of the images, it has potentials for analyzing skin facial features, such as facial hairs, smoothness, speckles, scars, wrinkles in real time. More functionality will be added on in the future. The software tool can work on a variety of different types of skin images, such as skin digital images, skin capacitive images, skin ultrasound images and more. Our study shows the apart from skin images, it can also work on other types of images, such as chest X-ray images. We have made the program available on GitHub so that others can benefit from it.

For the future work, we would like to test the skin image analysis tool on larger skin image dataset, on more volunteers, of different ages, genders, races and colors, as well as different skin conditions (dry, normal, silky, rough, etc.) and different skin diseases/cancers. For image classifications, more deep-learning neural networks will be implemented, and users would be able to select different training parameters. For skin image retrieval, users would be able to specify different parameter values for Gabor wavelet transform, PCA and GLCM. For live image analysis, more functions, such as image segmentation, object detections and so on, will be implemented.

**Author Contributions:** Conceptualization, P.X. and D.C.; methodology, P.X., X.O., X.Z., W.P., C.B. and E.C.; formal analysis, P.X., X.O., X.Z., W.P. and C.B.; investigation, X.O., X.Z., W.P. and E.C.; data curation, P.X. and C.B.; original draft preparation, P.X.; review and editing of manuscript, P.X. and D.C.; supervision, P.X. and D.C. All authors have read and agreed to the published version of the manuscript.

**Funding:** This research received no external funding.

**Acknowledgments:** We thank the London South Bank University and Biox Systems, Ltd., UK for the financial support. We also thank Longport, Inc., USA for loaning the EPISCAN I-200 instrument.

**Conflicts of Interest:** The authors declare no conflict of interest.

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
