# Peer review of "The Development of a Skin Image Analysis Tool by Using Machine Learning Algorithms"

_cosmetics, doi:10.3390/cosmetics7030067_

Round 1

Reviewer 1 Report

In their manuscript, Xiao et al. describe a skin image analysis tool by using machine learning algorithms. They already published a tool for skin analysis involving epsilon permittivity and show here that machine-learning tools can be used to improve the outcome of skin analysis software tools. This is interesting and will be a valuable tool for dermatologists doing patients care and also for research issues. I have some questions and suggestions for the authors:

Reference 8 and 9 are conference abstracts, it would be better to cite peer reviewed papers here

How many images have been used to test the training approaches?

Figure 5 and 6 are too small, it is difficult to see what is depicted in the screen shots

For most of the tests the authors state that “AlexNet has the shortest training time and ResNet101 has more accuracy”. Can the authors now suggest the “best” or the most suitable algorithms for skin image analysis? A clear statement is missing in the conclusion section.

Overall, the manuscript shall be revised for English language and for writing issues. In all different subsections of the results part the same sentences are used.

Author Response

We have revised the manuscript according to your comments, see attached file.

Reviewer 2 Report

I think that the topic is of interest, but the limitations are not mentioned and discussed in a profound way! I think that it is very imortant to mention also the limitations of a new method or technique! Therefore, I think that it is very important to state these limitations within the discussion part. But not only the limitations of the techniques, but also the limitation of the study itself. Furthermore, it is very important to give a small overview about the next requested steps! This is missing in the manuscript!!

Author Response

Discussions of limitations and future steps are added in revised manuscript.

Reviewer 3 Report

This is a not typical manuscript, something between a regular scientific paper and an advertisement. However, it is very interesting and presents results of original research. The “Conflict of interest” section is lacking, while it could be important in this particular case.

The text is well written but there are problems with numbering of figures. There are references to Figures 7 and 8 while there are only 6 figures in the manuscript.

Author Response

(The authors gave the same response as above.)

Round 2

Reviewer 2 Report

No further requests